# Relationship between Saliva and Sublingual Immunotherapy

**DOI:** 10.3390/pathogens10111358

**Published:** 2021-10-21

**Authors:** Aiko Oka, Mitsuhiro Okano

**Affiliations:** Department of Otorhinolaryngology, Head and Neck Surgery, International University of Health and Welfare Narita Hospital, 852 Hatakeda, Narita, Chiba Prefecture 286-0124, Japan

**Keywords:** sublingual immunotherapy, allergic rhinitis, microbiome, IgG4, saliva

## Abstract

The demand for allergen specific immunotherapy (AIT), especially sublingual immunotherapy (SLIT), is increasing because of its efficacy in inducing clinical remission of allergic diseases and its low risk of side effects. Since not all patients that undergo SLIT demonstrate an improvement in allergic symptoms, the development of biomarkers to predict the outcome and adjuvants for SLIT is desired. Saliva is the first target with which tablets used in SLIT come into contact, and salivary pH, chemical properties or microbiome composition are reported to possibly be associated with the outcome of SLIT. Antibodies such as IgG4 and IgA not only in the serum but also in the saliva are increased after SLIT and may also be associated with the efficacy of SLIT. The development of the metagenomic sequencing technique makes it possible to determine the microbiome composition and ratio of each bacterium, and researchers can investigate the relationships between specific bacteria and the immune response. Some bacteria are reported to improve the SLIT outcome and have the potential to be used as biomarkers for the selection of patients and as adjuvants in SLIT. Here, we introduce biomarkers for SLIT and present recent findings regarding the relationship between saliva and SLIT.

## 1. Introduction

Allergen-specific immunotherapy (AIT) can not only reduce allergic symptoms but also induce the clinical remission of patients from IgE-mediated allergic diseases including allergic rhinitis (AR), atopic asthma and venom allergy [1,2,3]. AIT consists of several routes of administration such as subcutaneous immunotherapy (SCIT) and sublingual immunotherapy (SLIT), and the demand for SLIT is increasing rapidly because it has fewer adverse effects and offers more convenience for patients than SCIT. The effectiveness of SLIT is reported to be similar to that of SCIT, but not all patients who undergo AIT experience substantial improvement in their allergic symptoms [4]. Biomarkers to predict or monitor the efficacy of SLIT have been investigated. Saliva may be important in SLIT because tablets used in SLIT first come into contact and react with the saliva in the sublingual site. Since a saliva sample is easy to collect without any pain for patients, it would be useful if there was evidence to support its potential to become a biomarker of SLIT. Saliva contains various microorganisms and establishes a local microbiome, and patients with not only local disorders such as dental caries or periodontal diseases but also systemic disorders such as allergic, inflammatory or malignant diseases are reported to have dysbiosis in their saliva [5,6,7]. The salivary microbiome may have some effects on the immune system and also on SLIT. According to the development of research on the salivary microbiome, the discovery of microorganisms which can be used as biomarkers to predict the effectiveness of SLIT and of probiotics to improve the outcome of SLIT is expected.

## 2. Biomarkers for Predicting the Efficacy of SLIT

Since AIT, including SLIT, needs a long-term treatment period accompanied by payment for treatment, methods to select treatment responder patients are desirable. Various biomarkers from blood samples have been investigated recently, but prognostic biomarkers collected ‘before’ SLIT are limited. Although elevated serum-specific IgE accompanied by allergic symptoms is needed to start SLIT, a clear association has not been shown between the pre-treatment-specific IgE level and clinical response. Some reports show that serum-specific IgE and the specific IgE-to-total IgE ratio before treatment can be prognostic biomarkers [8,9], but others show no evidence to support serum IgE as a prognostic biomarker [10,11]. Other immunoglobulins such as IgA and IgG are not reported to be pre-SLIT prognostic biomarkers. Pre-SLIT serum periostin, which is induced by type 2 cytokines in inflammatory diseases, is also reported as a biomarker for predicting SLIT in allergic rhinitis or asthma [11,12]. Baseline levels of IL-10 and IL-35, both of which are produced by Treg cells, are shown to be significantly correlated with the clinical efficacy of SLIT [8]. Serum metabolomics analysis shows that metabolites including lactic acid, ornithine, linolenic acid, creatinine, arachidonic acid and sphingosine can predict the efficacy of SLIT in allergic rhinitis [13]. These metabolic biomarkers suggest the contribution of metabolic pathways to the mechanisms of SLIT. Other biomarkers from blood samples reflecting the activation of basophils, cytokines, chemokines, Treg cells or Breg cells are unlikely to be able to predict clinical response to immunotherapy [14].

A body mass index over 25 before treatment is also reported to be associated with poor prognosis of SLIT in allergic rhinitis [10]. This result may reflect the insufficient doses of the allergens compared to the body weight of the patients, while obesity is also reported to be associated with poor control of asthma. Other in vivo biomarkers such as allergen provocation tests or chamber studies are unlikely to be able to predict the clinical response to immunotherapy [14].

## 3. Biomarkers for Monitoring the Efficacy of SLIT

Antigen-specific serum IgE is high in allergic rhinitis patients, and other antigen-specific antibodies such as IgG, IgG4 and IgA have been investigated recently. Serum and salivary IgG4 is reported to be lower in allergic patients than in non-allergic patients [15], although some other reports showed that antigen-specific serum IgG4 is higher in allergic rhinitis or eosinophilic chronic rhinosinusitis [16,17]. On the other hand, serum, bronchial secretary and salivary IgA is reported to be lower in allergic patients than in non-allergic patients [15,18,19]. However, both serum/salivary IgG4 and IgA are increased after SLIT in food allergy or allergic rhinitis patients [20,21,22]. IgG4 is induced along with IgE induction in response to allergen stimulation [23] and is shown to reduce inflammation as a result of inhibiting basophil activation or interfering with the immune complex formation of IgE and allergens as a blocking antibody [24,25]. IgA is known to be secreted across the intestinal mucosa and to have the function of blocking toxins or pathogenic microorganisms at the epithelial surface of the intestine [26,27]. In this era of the SARS-CoV-2 pandemic, salivary IgA has been reported to be associated with SARS-CoV-2 disease severity [28]. IgA secretion was shown to be induced by IL-10, which is an anti-inflammatory cytokine produced in response to immunotherapy [29,30]. Increased salivary IgA is desirable for patients undergoing SLIT in terms of reducing adverse effects in the oral cavity. Salivary IgG4 and IgA are increased in response to SLIT, and they may be biomarkers to assess the effectiveness of SLIT. On the other hand, salivary IgE is not detected in some reports [15,31], although IgE is an important immunoglobulin in nasal discharge in the diagnosis of allergic rhinitis. Basophil activation is reflected by the expression of diamine oxidase (DAO), which decomposes histamine. Increased intracellular DAO-positive basophils and decreased basophil histamine release are detected after SLIT [32,33].

## 4. Effects of Salivary pH on Immunity and SLIT

Saliva’s chemical properties may affect the immune system in the oral cavity. Saliva’s normal pH range is reported to be between 6.0 and 8.0 in healthy young adults [34]. Saccharolytic bacteria such as certain Streptococcus, Actinomyces and Lactobacillus species generate organic acids [35,36]. Arginine deiminase pathway (ADP)-positive bacteria such as certain species of Streptococcus and Lactobacillus maintain pH homeostasis in the oral cavity through alkali production [37,38]. The low reactivity of salivary IgA with Candida albicans cells grown at acidic pH values has been reported [39]. Another report showed that uric acid, the main antioxidant of saliva, and also salivary pH were significantly lower in HIV-infected individuals [40]. These results suggest that saliva with an acidic pH is associated with immune depression. Protein release from peanut flour at various pH values of saliva-like buffer was investigated, and the result showed the minimum extraction of proteins between pH 3 and pH 6, while protein extractability was high at both lower and higher pH values [41,42,43]. These results suggest that saliva with an acidic pH leads to poor solubility of allergen proteins used in SLIT. However, an association between salivary pH and the efficacy of SLIT has not been reported, and future investigation is desirable.

## 5. Salivary Microbiome and Allergic Diseases

The oral cavity is exposed to a huge variety of commensal and pathogenic microbiomes. High diversity is seen in a healthy microbiome, and an imbalance in the composition of the microbiome, called dysbiosis, leads not only to local disorders, such as dental caries, periodontal disease and oral cancer, but also systemic disorders such as allergic diseases, autoimmune diseases, rheumatic diseases and chronic bowel diseases [44,45,46,47]. The development of the metagenomic sequencing technique makes it possible to determine the microbiome composition and ratio of each bacterium, and researchers can investigate the relationships between specific bacteria and diseases. Children with asthma are reported to have lower diversity in their salivary microbiome, in addition to an increased abundance of Gemella haemolysans during early infancy, while an increased abundance of Lactobacillus species has been observed in healthy children [48]. The composition ratio of *Fusobacterium unclassified* and *Prevotella_6 unclassified* is lower and the composition ratio of *Fusobacterium nucleatum* is higher in the dental biofilm of asthmatic or atopic children than in that of healthy controls [49]. These reports suggest that dysbiosis in childhood affects the development of allergic diseases. However, the relationship between the salivary microbiome and the development of allergic diseases has been reported in a small number of studies, and further investigation would be desirable. Some reports show an association between the microbiome and allergic reactions. Allergens’ epitope similarity to microbiome sequences has been shown to be associated with low immunogenicity of allergens [50]. Extracellular vesicles (EVs) derived from the host microbiome have been shown to affect host immunity through stimulating host immune cells [51]. The EV is a membrane-enclosed vesicle, and the EV produced by Gram-negative bacteria contains lipids, outer membrane proteins and lipoproteins in addition to periplasmic and cytoplasmic components such as DNA and ATP [52,53]. For instance, the EV from *Porphyromonas gingivalis*, which is a Gram-negative oral anaerobe, is shown to activate NF-κB downstream of TLR signaling, resulting in the production of pro-inflammatory cytokines (TNF, IL-8 and IL-1β). On the other hand, the EV from *P. gingivalis* is also shown to induce the anti-inflammatory cytokine IL-10 [54].

## 6. Salivary Microbiome and SLIT

Human monocytes are reported to produce IL-10 in response to saliva, and IL-10 production is associated with signals through TLR2 and TLR4 [55]. These results show that saliva which contains the oral microbiome is highly associated with immunity. On the other hand, salivary IgG4 and IgA are increased in response to SLIT, as shown above. Because the tablet used in SLIT first comes into contact with the oral cavity or saliva, the salivary microbiome may have some effects on SLIT. With the use of the metagenomic sequencing technique, organisms in the salivary microbiome of Japanese cedar-allergic rhinitis patients were shown to be mainly composed of *Firmicutes* (median: 38.4%), *Bacteroidetes* (median: 16.9%) and *Proteobacteria* (median: 14.4%). On the other hand, in non-allergic controls, the composition ratios of *Firmicutes*, *Bacteroidetes* and *Proteobacteria* were 31.8%, 9.6% and 37.3%, respectively. The composition ratio of *Bacteroidetes* in the saliva was significantly higher compared to the non-allergic controls, and the composition ratio of *Bacteroidetes*, especially *Prevotella* species, was also shown to have a positive correlation with IL-10 production in cedar pollen-allergic patients. The composition ratios of *Bacteroidetes* and *Prevotella* were also significantly higher in asymptomatic patients showing a visual analog scale (VAS) of 0 after SLIT compared with symptomatic patients showing a VAS above 0 after SLIT [56]. VAS is well validated for the measurement of AR symptoms and correlates well with allergic rhinitis and its impact on asthma (ARIA) severity classification; it also correlates well with the reflective Total Nasal Symptom Score (rTNSS) and rhinoconjunctivitis quality of life questionnaire (RQLQ) [57]. These results suggest that *Prevotella* in the salivary microbiome may have some effects in inducing IL-10 production and in reducing allergic symptoms (Figure 1), and it may be able to be used as an adjuvant for SLIT. The *Lactobacillus* genus is commonly reported as probiotics to engage receptors such as TLR2 and TLR4 [58]. Some *Lactobacillus* species work as adjuvants in SLIT experiments using mice with asthma [59,60]. On the other hand, certain Lactobacillus species increase the antibody responses to an allergen after it is used as a probiotic in SLIT [61]. Since not all allergic patients benefit from SLIT, and also due to the fact that SLIT takes a long time to express the effect of symptom reduction, these microorganisms improving the efficacy of SLIT may potentially be used as adjuvants for SLIT. In addition, SLIT may have the potential to alter the human microbiome, although we found one clinical report showing that the nasal microbiome was not significantly different between AR patients with and without allergen immunotherapy [62]. We would like to investigate the difference between the salivary microbiome in AR patients before and after SLIT in addition to healthy controls. Further research regarding the relationship between immunotherapy and the microbiome is needed.

## 7. Conclusions

The demand for AIT, especially SLIT, is increasing because of its efficacy in inducing clinical remission of allergic diseases and its low risk of side effects. Since not all patients that undergo SLIT demonstrate an improvement in allergic symptoms, biomarkers to predict the outcome and adjuvants for SLIT are expected. Saliva is the first target with which tablets used in SLIT come into contact, and antibodies in saliva such as IgG4 and IgA are increased after SLIT. The salivary microbiome is reported to have some effects on the immune system and is associated with allergic and immune diseases. According to the development of the metagenomic sequencing technique, specific microorganisms which turn out to be associated with the efficacy of SLIT can be detected, and these microorganisms may be used as adjuvants for SLIT in the future.

## Figures and Tables

**Figure 1 pathogens-10-01358-f001:**
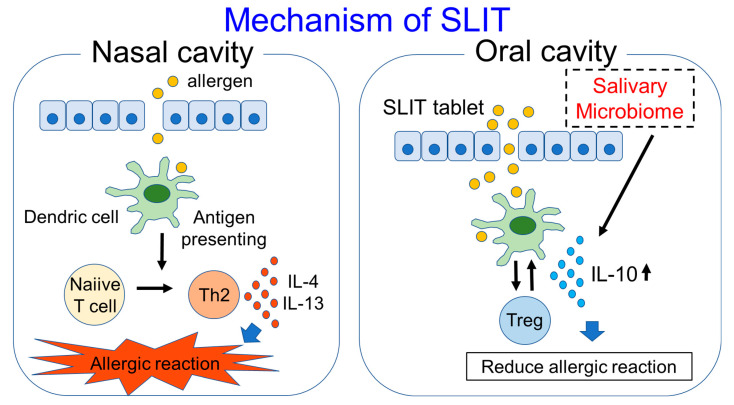
Mechanism of SLIT and participation of salivary microbiome in SLIT are shown in this figure.

## Data Availability

No new data were created or analyzed in this study. Data sharing is not applicable to this article.

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
