# Peer review of "Relationship between Saliva and Sublingual Immunotherapy"

_pathogens, 2021, doi:10.3390/pathogens10111358_

Round 1

Reviewer 1 Report

The authors wrote a comprehensive review about the relationship between saliva and sublingual immunotherapy (SLIT). Since the current understanding and progress of the research regarding immunoglobulin reaction and salivary microbiome in relation to SLIT was described with referring a series of recent reports including their own findings, this review is promising to provide useful information to various readers working on allergy and immunotherapy fields. The reviewer suggests several minor modifications.

  1. Including recent findings of biomarker research for SLIT using biological samples other than saliva would enrich the value of this study.
  2. Including a couple of topics regarding examinations for SARS-CoV-2 infection using salivary samples may attract more readers to this manuscript.
  3. Line 99. Since the control data of non-allergic patients were not included, Figure 2 did not show the high composition ratio of Bacteroidetes in saliva in cedar pollen allergic patients. This figure also did not indicate the high ability to produce IL-10.
  4. Line 102. The reviewer recommends to including the explanation for a visual analog scale (VAS).
  5. Changing the term “antigen” to “allergen” is desirable.
  6. Performing additional English editing may be required.

Reviewer 2 Report

The review entitled Relationship between saliva and sublingual immunotherapy aims at revising the current literature on how saliva, and saliva microbiome in particular, impacts SLIT.  I have detected the following limitations:

  • The structure of the text is not articulate. It is not clear what is the aim/scope of the review or what it contributes to the current literature. Please, try to reflect on this and define a clear aim.
  • When the aim is to summarize/revise the current knowledge on the impact of saliva on SLIT, every component of saliva should be described and taken into account (e.g. pH, chemical composition, microbiota composition) to assess its impact. Both positive and negative outcomes should be described (e.g. a high pH facilitates SLIT/ a specific microbial composition makes SLIT less effective- please, note these are just examples. Please, check into the literature for accurate information).
  • Since the title of the review is broad, you should explore the reciprocal relationship between saliva and SLIT, i.e. how saliva composition affects SLIT and how SLIT may alter saliva composition. Explore what will be the impact of this relationship on the disease outcome.
  • Regarding the described diseases, the text jumps from one to another without a clear reasoning. Please try to focus the review on just one allergic disease (e.g. allergic rhinosinusitis) or make it comprehensive and systematic by revising saliva and SLIT relationship in several allergic disorders.
  • For a review, the text is very short and evasively informative. Please, include the changes suggested above and expand the information on the topic accordingly. Consequently, more references should be added supporting the information. Please, try to avoid much self-citation.
  • A graphical abstract or schematic figure would also add value to the review, instead of including previously published figures extracted from autocitations.
  • Finally, the readability of the text in English would profit from the revision of an English native speaker once the scientific content of the manuscript is edited.

Reviewer 3 Report

The authors revealed what we know on the topic of microbiome and immune system interaction, related to SLIT. Since the mutual relationship of the former two is well-established, the role of SLIT in it has to be elucidated. 

The article is consistent within itself. The references are relevant and recent. The cited sources are referenced correctly. Appropriate and key studies are included. The paper is comprehensive, the flow is logical and the data is presented critically.
However, there are some specific comments on weaknesses of the article and what could be improved.
Specific comments on weaknesses of the article and what could be improved:
Major points - none
Minor points

1. Please, comment if here are any clinical trials that researched the effects of SLIT on human microbiome and/or extended your hypothesis how these effects could be tested, and what design you will suggest.

Round 2

Reviewer 2 Report

Although the readibility of the text has improved in general terms, the aim and content of this review is still not clear since, in my opinion, the text lacks of a coherent structure. Therefore, it is difficult to assess its contribution to the field in its current form. New references supporting the new information included have been added.

  • What is the difference between "Biomarkers for predicting efficacy of SLIT" and "Biomarkers of response after SLIT"? You include the following information (lines 88-90) "Salivary IgG4 and IgA are increased in response to SLIT, and they may be biomarkers to assess the effectiveness of SLIT." under the heading "Biomarkers of response after SLIT". Are then salivary IgG4 and IgA predictive biomarkers of efficacy? Please, try to organise the information to avoid repetition and send a clearer message.
  • In your response to the reviewer, you state that a graphical abstract was included in the revised version but I have not found it.
  • I suggest again that you remove the figures adapted from reference 33, since they do not add any value to the review and can be found in the original citation.
  • Some English editing is still required in some parts of the text. The text would also benefit from avoiding repetition in the abstract, introduction and conclusions sections.

Round 3

Reviewer 2 Report

I have no further suggestions